# The Promising Potential of Triploidy in Date Palm (*Phoenix dactylifera* L.) Breeding

**DOI:** 10.3390/plants13060815

**Published:** 2024-03-12

**Authors:** Ahmed Othmani, Hammadi Hamza, Karim Kadri, Amel Sellemi, Leen Leus, Stefaan P. O. Werbrouck

**Affiliations:** 1Laboratory for In Vitro Tissue Culture, Regional Centre for Research in Oasis Agriculture, Tozeur Km1, Degueche 2260, Tunisia; degletbey@yahoo.fr (A.O.); amelsallami1@gmail.com (A.S.); 2LR21AGR03-Production and Protection for Sustainable Horticulture (2-PHD), Regional Research Centre on Horticulture and Organic Agriculture Chott Mariem, University of Sousse, Sousse 4042, Tunisia; 3Arid and Oasis Cropping Laboratory, Institute of Arid Lands, Medenine 4119, Tunisia; hamzapalmier@yahoo.fr; 4Biotechnology and Genetic Resources Laboratory, Regional Centre for Research in Oasis Agriculture, BO 62, Degueche 2260, Tunisia; kadrikarim2001@yahoo.fr; 5Laboratory of Biotechnology Applied to Agriculture, National Institute for Agronomic Research of Tunis, University of Carthage Tunis, Ariana 2049, Tunisia; 6Plant Sciences Unit, Flanders Research Institute for Agriculture, Fisheries and Food (ILVO), Caritasstraat 39, 9090 Melle, Belgium; leen.leus@ilvo.vlaanderen.be; 7Department of Plant & Crops, Faculty of Bioscience Engineering, Ghent University, Valentin Vaerwyckweg 1, 9000 Ghent, Belgium

**Keywords:** female fertility, flow cytometry, hybridization, phenotype, triploid

## Abstract

Date palms are a vital part of oasis ecosystems and are an important source of income in arid and semi-arid areas. Crossbreeding is limited due to the long juvenile stage of date palms and their dioecious nature. The aim of this study was to create triploid date palms to obtain larger and seedless fruits and to increase resilience to abiotic stresses. A tetraploid date palm mutant was crossed with a diploid male palm, yielding hundreds of seeds suspected of containing triploid embryos. Six years after planting, four palms with confirmed triploidy reached maturity. They are phenotypically distinct from diploids, with a thicker rachis, thinner spines, wider and longer midleaf spines, and a longer apical spine. They were classified as sterile bisexual, sterile male and fertile female. One of the latter produced very tasty dates with a very small seed, which is promising for the marketability and profitability of date palm fruits. This first report on triploid date palms provides a way in which to make a significant leap forward in date palm breeding. Given the vigor and fruit quality of female triploid date palms, compared to their diploid counterparts, they will be the target of breeding programs and may spearhead new oases.

## 1. Introduction

In the arid and semi-arid areas of North Africa and the Middle East, date palms (*Phoenix dactylifera* L.) are the cornerstone of oasis ecosystems and serve as the primary income source. However, the sustainability and profitability of this desert staple are under threat from various biotic and abiotic stressors, with climate change due to global warming being the most significant [1,2]. Enhancing the resilience of date palms to stressors and securing food supply can be achieved through the development of genetic variants. However, traditional crossbreeding is hindered by the extended juvenile phase of date palms, their dioecious nature, and the absence of early selection for most agronomic traits [3]. While molecular tools such as marker-assisted selection (MAS), genome editing (GE), genome-wide association studies (GWASs), and a comprehensive set of single nucleotide polymorphisms (SNPs) offer potential solutions, their application in palms is costly and time-intensive [2].

The creation of ploidy variants presents a viable strategy for enhancing phenotypic variation. A tetraploid was discovered in a lateral shoot of ‘Deglet Nour’ in Tozeur, Tunisia [4]. While triploids have not yet been produced in date palms, they have proven to be more robust in other fruit crops, yielding larger fruits with minimal or no seeds. Triploids can be generated through (1) crossbreeding between tetraploid and diploid cultivars, (2) cross-pollination with (artificially induced) 2n gametes, (3) in vitro regeneration of triploid endosperms via adventitious shoots, and (4) the fusion of haploid and diploid cells in protoplasts. This report utilized crosses between the tetraploid shoot of ‘Deglet Nour’ and diploid male genotypes. Although triploids have recently been reported in the African oil palm by Pomiès et al. [5], to our knowledge, this is the first report on triploid date palms. This study provides a way to leap forward in date palm breeding in a straightforward manner.

## 2. Results

### 2.1. Hybridization between Tetraploid and Diploid Staminate Date Palm (P3)

Tetraploid flowers of ‘Deglet Nour’ offshoots, when pollinated by the diploid male ‘P3’, yielded fruits that were not only edible but also heavier and larger in volume compared to the dates produced by diploid flowers (Figure 1). The majority of the fruits obtained were parthenocarpic. Only a small fraction, specifically 0.9%, of these fruits formed seeds that effectively gave rise to a triploid plant (Table 1).

### 2.2. Analysis of the Ploidy Level of the Four Plants with Flow Cytometry

As depicted in Figure 2, the obtained histograms each display a peak with a value of approximately 100. This value corresponds to the nuclear DNA of the diploid control ‘Deglet Nour’. Additionally, there is a prominent peak with a relative value of 150, which is attributed to the triploid nuclear DNA of the plants under examination.

### 2.3. Leaf Morphology

The four triploid genotypes exhibited phenotypic differences when compared to their diploid counterparts. These differences included a thicker rachis, thinner spines, and wider and longer midleaf spines (Appendix A; Figure 3). Additionally, they had a longer apical spine. Except for T1, all triploid plants had, on average, longer leaves, a smaller maximum length of spines, and a larger number of leaflets per leaf than the standard diploid plants. However, when parameters such as leaf width at the base of the petiole, average number of spines, and upper spine width were considered, it was not possible to distinguish the triploids from the diploid control plants (Appendix A). Although the triploids had a more vigorous appearance, no difference was observed in overall plant length or width between the triploids and diploids.

### 2.4. Reproductive Properties

In 2021 and 2022, the four triploid plants reached maturity. As shown in Table 2, compared to the diploid control ‘Deglet Nour’, the inflorescences of these plants emerged later and their fruits took additional weeks to ripen. In 2021, the inflorescences of the triploid hybrids appeared a month or more later than the diploid hybrids. T4 did not flower in 2021, but it did for the first time in 2022. 

T1 (Figure 4A) had bisexual flowers with six stamens and three carpels (Figure 4B–D). These flowers were likely female sterile, as they only produced parthenocarpic fruits after being pollinated with T3 pollen for three consecutive years (Figure 4E,F). The plant was also male sterile, as the pollen did not germinate, although 10% of it was viable (Figure 4G–I). T2 (Figure 5A,B) was male because this plant only produced flowers with six stamens each (Figure 5C,D). These stamens were sterile, similar to those of T1. The stamens eventually degenerated in the same manner as those of normal male (staminate) date palms (Figure 5E). Unlike T1 and T2, T3 and T4 appeared to be typical female (pistillate) date palms, with flowers consisting of three carpels and six aborted stamens (Figure 6 and Figure 7).

Following pollination and fertilization with standard diploid pollen, the flowers of T3 and T4 yielded edible fruits that contained seeds. Both T3 and T4 produced fruits that exhibited an extremely low seed-to-fruit weight ratio (Figure 6E,F). This ratio was notably lower compared to ‘Deglet Nour’ and even other commercial varieties such as ‘Mejhool’ and ‘Berhi’ that were cultivated at the same location (Table 3 and Figure 8). Interestingly, the seeds of T4 were capable of germination, unlike those of T3 (Figure 9).

## 3. Discussion

Only four triploid date palm genotypes were successfully produced by pollinating tetraploid flowers with diploid pollen. So, the success rate was remarkably low, with only 0.9% of fruits containing seeds in both 2014 and 2015, followed by a slight increase to 1.05% in 2016. These findings suggest that tetraploid date palms exhibit a high degree of post-zygotic reproductive isolation from their diploid progenitors. This reproductive barrier, known as the triploid block, is attributed to abnormalities in the growth and structure of the endosperm [6,7], which in this case is likely pentaploid. According to the concept of the endosperm balance number (EBN), which states that the hybrid endosperm needs a 2:1 maternal to paternal ratio for normal development, this cross has a problematic 4:1 ratio, which may also explain the high number of failing offspring [8].

As has been observed in other plant species [9,10,11,12], we were able to differentiate triploid date palm trees from their diploid counterparts based on specific characteristics of their leaves and fruits. Generally, polyploidy can lead to an increase in the size of cells, organs, or even the overall habitus [13]. The presence of three alleles per gene could account for the potential high gene dosage effects and strong heterosis observed in triploid selections [14].

Within the palm family, the oil palm serves as the only reference. Pomiès et al. [5] reported that spontaneously induced triploid African oil palms exhibit greater vigor, thicker petioles, thicker stems, and a larger habit than diploid palms. They also noted that polyploidy impacts phyllotaxy. According to Van de Peer et al. [12,15], the competitive edge of triploids over their diploid offspring is primarily due to their ability to capture a relatively larger amount of photosynthetic energy through their larger leaves.

Theoretical models in the literature often posit that triploids typically exhibit reduced fertility or even complete sterility [16,17]. Contrary to this assumption, we observed that T4 was capable of producing fruits with germinating seeds. However, further cytological studies are needed to determine the extent to which the embryos are aneuploid due to the formation of multivalents and other meiotic defects. In contrast, none of the small seeds of T3 were able to germinate. This could be attributed to the embryos being aneuploid and/or the seed having insufficient reserves to nourish the embryo during the initial stages of germination. These observations suggest the potential for employing in vitro techniques to rescue the embryos.

T1 produced bisexual flowers that contained both non-functional male and female organs. This type of flower was previously described by [18] in two aberrant female date palm variants of the cultivar ‘Alligue’. Cherif et al. [19] demonstrated that date palms possess an XY chromosome system, which is considered to be among the oldest sex chromosomes in flowering plants [20]. Consequently, the occurrence of bisexual flowers may be due to the presence of an additional Y chromosome alongside the normal XX set found in female plants. While this has not yet been genetically confirmed, it suggests that the tetraploid mother plant has an XXXX configuration, which resulted in XXY (T1 and T2) and XXX (T3 and T4) offspring after cross-pollination with an XY male.

In terms of size and weight, the fruits of T3 and T4 are between those of the diploid and tetraploid ‘Deglet Nourdi’. The fruits of the diploid ‘Mejhool’ take the crown as the largest, while those of the diploid ‘Barhi’ are the smallest (Table 3). Although the female triploids T3 and T4 do not differ much in their general morphology, they show much difference (Table 3) in fruit size. Triploid T3 produced large fruits with excellent organoleptic quality and a very low seed/fruit weight ratio, which could be of great commercial interest. Triploid T4, on the other hand, had an almost normal seed size, while the fruits were not particularly palatable. The discovery of T3 paves the way for screening many more triploid date palms with comparable horticulturally interesting traits. The triploids flowered later and their fruits, if present, ripened later, which is a common trait in polyploid plants [19]. This provides them with the advantage of avoiding both the recently occurring extremely hot temperatures of August–September and the attacks of date moths that are active in these months. Firmer leaves are also thought to be more resistant to a number of foliar diseases, a hypothesis that needs to be assessed over several years. T1 produced several inedible seedless fruits via parthenocarpy, which, like the parthenocarpy fruits sometimes observed in ‘Deglet Nour’, can be exploited for their chemical, functional and technological properties [21]. 

## 4. Materials and Methods

### 4.1. Hybridization between Tetraploid and Diploid Date Palms

Pollen was harvested from the common male diploid cultivar ‘P3’ in the month of February across the years 2015, 2016, and 2017. This pollen was then used to pollinate manually opened mature female inflorescences of the tetraploid lateral shoot of a chimeric ‘Deglet Nour’. To prevent random mating, the flowers were promptly bagged post-pollination. These bags were removed after a period of 15 days. The late-flowering staminate date palm ‘P3’ was chosen for this process due to its late flowering characteristic, similar to the tetraploid date palm. In addition, the pollen of ‘P3’ was used to pollinate flowers of the diploid sector of the chimeric date palm. This process was completed to produce diploid control fruits and seeds.

### 4.2. Germination and Plant Care

Seeds were extracted from ripe fruits 5 to 6 months post-pollination and thoroughly cleaned to remove all pulp. These seeds were then rinsed in 90% (*v*/*v*) ethanol and placed on a damp cloth within a sealed plastic box. This box was kept in an incubator at a temperature of 26 ± 2 °C in complete darkness. After a month of incubation, germinated seeds were observed. They were then transferred to pots filled with a 1:1 mixture of sand and peat moss. The pots were moved to a greenhouse where they were exposed to natural sunlight and maintained at a temperature of 28 ± 2 °C with a relative humidity of 60–70%. Watering was performed manually twice a week using a watering can. Three months later, the plants were transplanted into the open field of the experimental garden of the Regional Research Center for Oasis Agriculture (CRRAO). Here, they were grown alongside diploid offshoots of ‘Deglet Nour’. The plants were watered via drip irrigation at a rate of 100 L per hour. The frequency of watering varied with the seasons—once a week in winter, twice a week in autumn and spring, and thrice a week in summer. Each plant was annually fertilized with 5 kg of sheep manure. The plants were maintained under these conditions for a period of seven years.

### 4.3. Analysis of Seedlings by Flow Cytometry

Ploidy analysis was performed by means of flow cytometry [22], using a CyFlow Space (Sysmex, Münster, Germany) flow cytometer equipped with a UV-light-emitting diode and Flomax 2.9 software (Quantum Analysis, Münster, Germany) to determine peak positions on the histograms obtained. Sample preparation consisted of chopping leaf tissue (according to Galbraith et al., 1983 [23]) and sample preparation including 4′,6-diamidino-2-phenylindole (Sigma-Aldrich, Overijse, Belgium) staining, according to Otto (1990) [24]. Standard leaf material from a diploid control plant of the variety ‘Deglet Nour’ was used as an external reference.

### 4.4. Pollination of Triploid Plants

The four obtained triploid hybrids were coded as T1, T2, T3 and T4. In 2021 and 2022, flowers from two pistillate hybrid plants (T3 and T4) were pollinated with pollen from ‘P3’ using the pollination method mentioned above. 

### 4.5. Morphological Observation and Measurements

The descriptors developed by IPGRI [25] for date palms were used to characterize the leaves of triploid and diploid control plants. The length of the palm (Lp), the thickness of the rachis (Tr), the width of the palm at the base of the petiole (Wp), the maximum. thickness of the spines (Tsp), the maximum length of the spines (Lsp), the maximum width of the spines in the midleaf (Wsp), the maximum length of the spines in the middle of the palm (Lsm), the length of the apical spine (Las), and the width of the apical spine (Was) were measured with a tape measure and caliper. The number of spines (Nsp) and the number of leaflets (NLf) were counted.

Seed and fruit weights were determined for thirty fruits from the genotypes T3 and T4 and compared to two commercial varieties ‘Mejhool’ and ‘Berhi’ and the original diploid ‘Deglet Nour’. All plants had the same age and were grown in the trial garden of the CRRAO.

### 4.6. Viability of Pollen

The viability of pollen was assessed in triplicate using three flowers. The pollen was placed on a slide with 1–2 drops of a 1% acetocarmine dye solution and covered with a coverslip. The slide was then examined under a microscope (Biological Microscope, Wuzhou New Found Instrument L1100A, Wuzhou, China) at 400× magnification. Pollen grains that exhibited red-stained cytoplasm were deemed viable. Pollen germination was also evaluated in triplicate, again using three flowers. The pollen grains were deposited onto the medium of Brewbaker and Kwack [26] in three separate Petri dishes. Brushes were used to ensure proper separation of the pollen. Following a 24 h incubation period at 28 °C [27], a square centimeter of the medium was excised and placed on a microscope slide for observation at 400× magnification. A pollen grain was considered to have successfully germinated if the length of the pollen tube was equal to or greater than the diameter of the grain [28].

### 4.7. Statistical Analysis

The results were expressed as the mean value ± standard deviation. The studied parameters were analyzed separately using ANOVA with post hoc Student–Newman–Keuls comparisons with SPSS 16.0. A confidence level of *p* = 0.05 was used to identify significant differences between experimental conditions.

## 5. Conclusions

In a pioneering effort, the phenotype of triploid date palms has been characterized using four completely different genotypes. The excellent flavor and small seeds of one of these female triploid date palms will be the focus of a breeding program. and could potentially lead to the establishment of new oases in regions experiencing escalating biotic and abiotic stress. This is due to their superior vigor and fruit quality compared to their diploid counterparts. The offshoots of the plants will be used for in vitro initiation and micropropagation. In the near future, these triploid date palms will also be used for detailed karyotyping and to detect genomic and transcriptomic variation in the near future.

## Figures and Tables

**Figure 1 plants-13-00815-f001:**
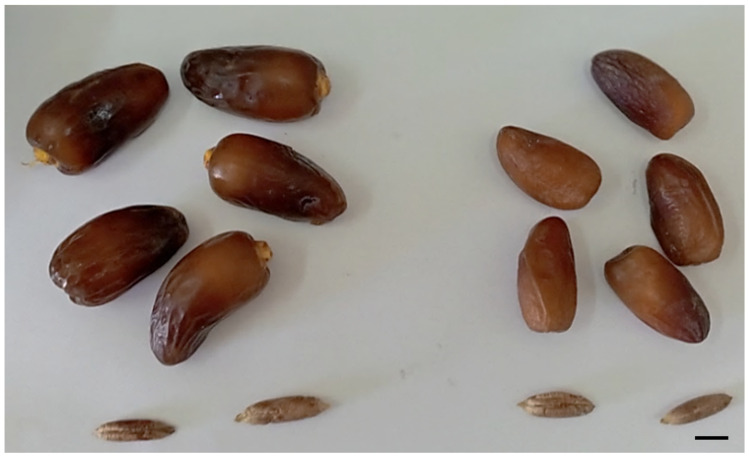
Edible fruits and seeds produced by tetraploid (**left**) and diploid (**right**) sectors of the chimeric date palm. Size bar = 1 cm.

**Figure 2 plants-13-00815-f002:**
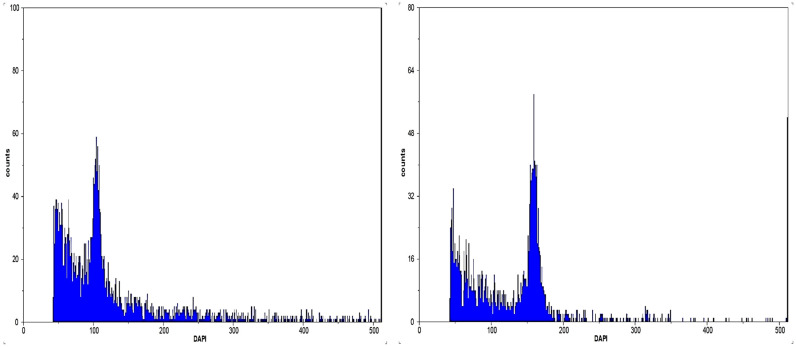
Histogram of DNA content of ‘Deglet Nour’ (2×) (**left**) and the triploid date palm (T3) (**right**), with the fluorescence intensity in the X-axis and number of nuclei in the Y-axis. Signal around fluorescence intensity 50 is noise.

**Figure 3 plants-13-00815-f003:**
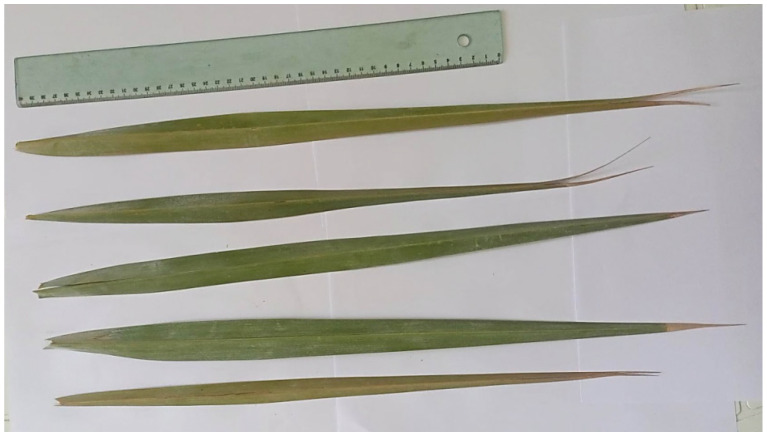
Mid-leaf leaves of ‘Deglet Nour’ (2×) and the triploid genotypes T1, T2, T3 and T4, from bottom to top, respectively.

**Figure 4 plants-13-00815-f004:**
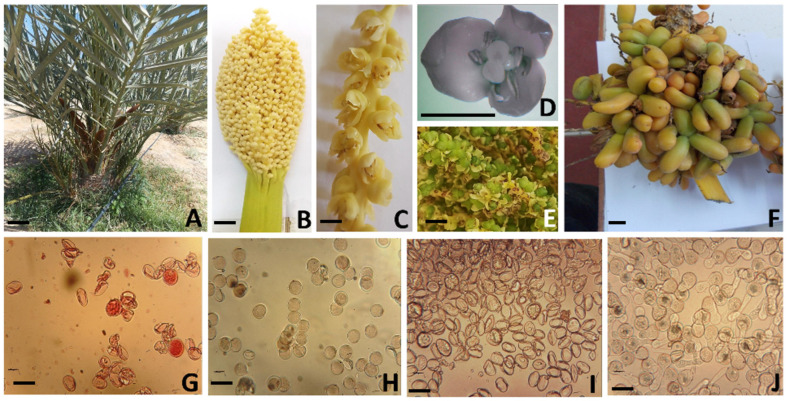
Morphological and reproductive characteristics of T1. (**A**) Overview of T1 with spathes; (**B**) inflorescence; (**C**) flower axis; (**D**) apparent bisexual flower; (**E**) flower development after 2 months of spathe dehiscence; (**F**) inedible parthenocarpic fruits after 6 months of spathe dehiscence (scale bar = 1 cm); (**G**,**H**) viability test of pollen of T1 and P3; respectively; (**I**,**J**) germination test of pollen of T1 and P3 (scale bar = 30 µm).

**Figure 5 plants-13-00815-f005:**
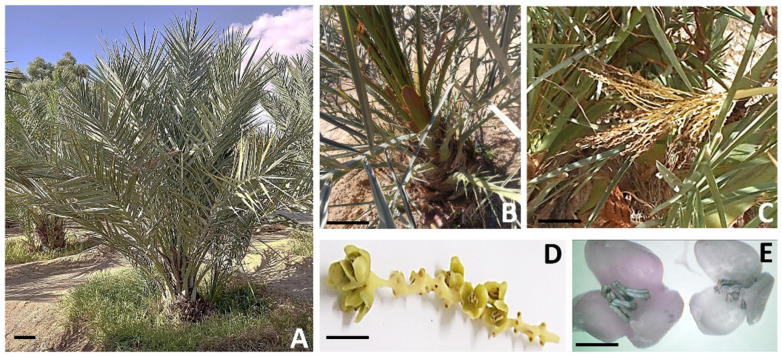
Morphology of vegetative and reproductive apparatus of T2. (**A**) Overview of T2 (scale bar = 10 cm); (**B**) overview of T2 with spathes; (**C**) degenerate inflorescence; (**D**) flower axis (spikelet of flowers); (**E**) flower of T2 (**left**) and male flower of P3 (**right**) (scale bar = 1 cm).

**Figure 6 plants-13-00815-f006:**
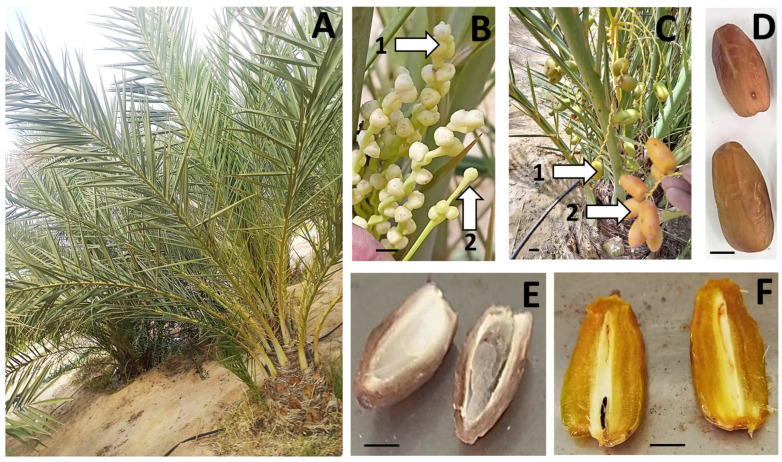
Morphology of vegetative and reproductive apparatus of T3. (**A**) Overview; (**B**) flower axis of T3 (1) and ‘Deglet Nour’ (2); (**C**) fruits of T3 at *khalal* stage (1) and ‘Deglet Nour’ at *rutab* stage (2); (**D**) ripe fruits (*tamer* stage) of ‘Deglet Nour’ (**top**) and T3 (**bottom**); (**E**) halved matured fruits of ‘Deglet Nour’; (**F**) halved matured fruits of T3, showing a very small black seed (scale bar = 1 cm).

**Figure 7 plants-13-00815-f007:**
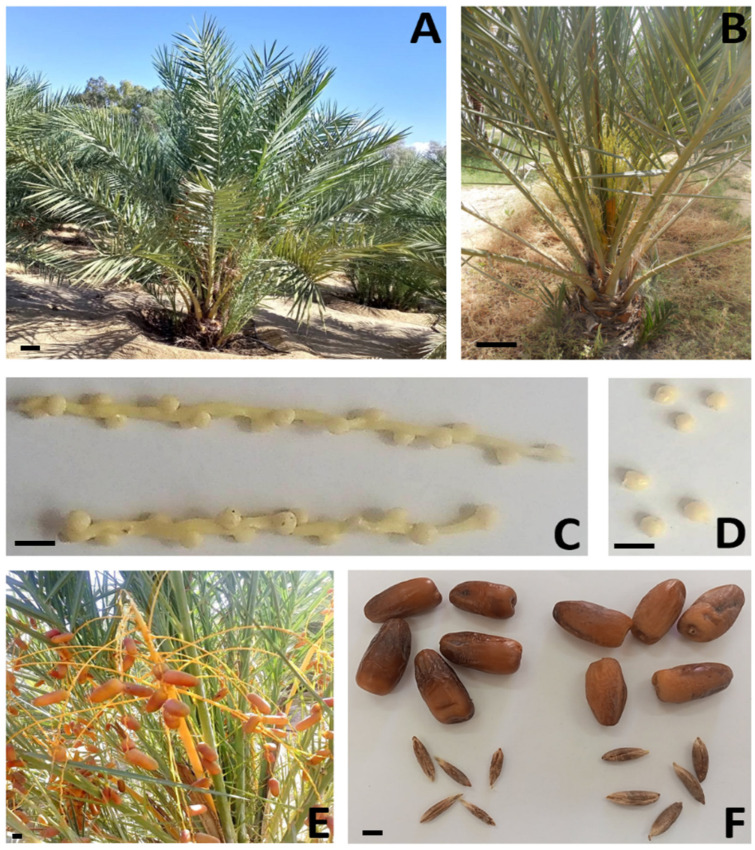
Morphology of vegetative and reproductive apparatus of T4. (**A**) Overview (scale bar = 10 cm); (**B**) overview inflorescences in plant; (**C**) flower axis of ‘Deglet Nour’ (**top**) and T4 (**bottom**) just after opening of spathes; (**D**) flowers of ‘Deglet Nour’ (**top**) flowers of T4 (**bottom**); (**E**) fruits of T4 in ‘*rutab*’ stage; (**F**) mature fruits (*tamer* stage) and seeds of ‘Deglet Nour’ (**right**) and T4 (**left**) (scale bar = 1 cm).

**Figure 8 plants-13-00815-f008:**
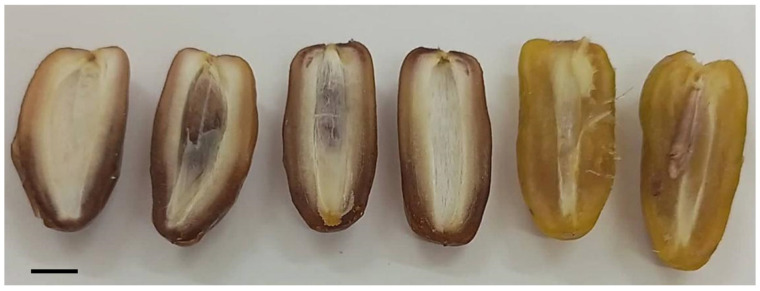
From left to right: halved dates of ‘Deglet Nour’, T4 and T3 (scale bar = 1 cm).

**Figure 9 plants-13-00815-f009:**
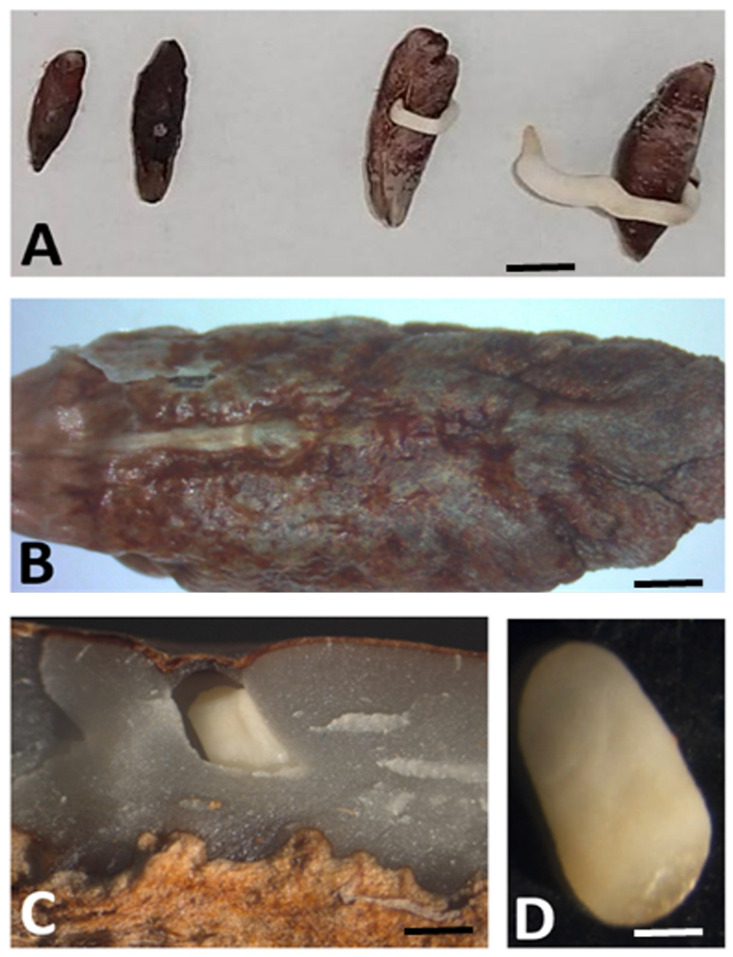
(**A**) Growth blockage of germinated embryos from T3 (**left**) and germinating embryos from T4 (**right**) (scale bar = 1 cm); (**B**) seed of T3 (scale bar = 2 mm); (**C**,**D**) embryo in T3 seed (scale bar = 0.5 and 0.2 mm).

**Table 1 plants-13-00815-t001:** Characteristics of fruits produced by the tetraploid ‘Deglet Nour’ shoot after crossing with P3 in 2014, 2015 and 2016. nf: number of fruits, nef: number of edible fruits, ef: edible fruits, nsf: number of seed fruits, sf: seeded fruits, nge: number of germinated embryos, gf: germination frequency, fpo: fruits producing offspring.

Year	Total nf	nef	ef (%)	Total nsf	sf (%)	nge	gf (%)	fpo (%)
2014	3420	56	1.63	45	1.31	31	68.88	0.9
2015	4100	68	1.65	57	1.39	39	68.42	0.9
2016	5004	98	1.95	74	1.47	53	71.62	1.05
Average	4175	74	1.74	59	1.39	41	69.64	0.95

**Table 2 plants-13-00815-t002:** Date of first inflorescence emergence and fruit maturation in seedlings of the diploid control ‘Deglet Nour’ (T0) and triploids (T1, T2, T3 and T4) during 2021–2023.

Year	Phenology	T0	T1	T2	T3	T4
2021	Inflorescence emergence	5 February	16 March	4 March	28 March	-
	Fruit maturation	7 September	-	-	18 September	-
2022	Inflorescence emergence	22 February	-	-	25 April	24 March
	Fruit maturation	16 September	-	-	28 September	21 September
2023	Inflorescence emergence	28 February	12 March	20 March	14 April	11 March
	Fruit maturation	3 September	-	-	21 October	28 September

**Table 3 plants-13-00815-t003:** Fruit weight, seed weight and ratio (seed weight/total weight) of the cultivars ‘Deglet Nour’ (DN), ‘Mejhoul’, ‘Barhi’, T3, T4 and the tetraploid offshoot (Tet) harvested from plants cultured in the CRRAO trial field. Means (±SD) followed by the same letter are not significantly different (Student–Newman–Keuls (S-N-K), *p* ≤ 0.05).

Genotype	Fruit Length (cm)	Fruit Width (cm)	Fruit Weight (g)	Seed Weight (g)	Seed Weight/Fruit Weight (%)
DN	3.83 ± 0.15 e	1.64 ± 0.10 e	8.59 ± 0.29 f	0.92 ± 0.12 b	10.64 ± 2.41 a
T3	4.26 ± 0.39 c	1.54 ± 0.19 f	10.76 ± 1.69 d	0.17 ± 0.09 e	1.53 ± 0.70 f
T4	4.02 ± 0.28 d	1.74 ± 0.23 d	9.60 ± 1.15 e	0.31 ± 0.13 d	3.11 ± 1.01 e
Tetra	4.50 ± 0.25 b	2.15 ± 0.20 c	18.13 ± 1.79 b	0.69 ± 0.20 c	3.80 ± 1.14 d
‘Mejhool’	4.82 ± 0.52 a	2.61 ± 0.11 a	23.33 ± 2.01 a	1.52 ± 0.20 a	6.50 ± 0.62 b
‘Barhi’	3.44 ± 0.21 f	2.43 ± 0.13 b	13.66 ± 1.10 c	0.68 ± 0.16 c	4.93 ± 1.03 c

## Data Availability

Data are available upon reasonable request.

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
