# Peer review of "The Promising Potential of Triploidy in Date Palm (Phoenix dactylifera L.) Breeding"

_plants, 2024, doi:10.3390/plants13060815_

Round 1

Reviewer 1 Report

Comments and Suggestions for Authors

Date palm is an important oasis ecological species and an important sugar fruit plant. In view of its biological and economic attributes, it is appropriate and effective to adopt a genetic improvement strategy of polyploid breeding. It is pleased to see that the research team has achieved good results, and hope that through continuous efforts, excellent varieties can be bred.

Overall, triploid breeding is successful for date palms, but what remains unresolved is, the yield of triploids produced by crossing tetraploids and diploids is very low, and the reproductive biology mechanism has not yet been fully revealed. This manuscript also does not essentially summarize methods for efficient breeding of triploid date palms. But what is beneficial is that the manuscript has characterized the fertility of triploid date palm in detail, which is very important for further developing date palm genetic improvement plans.

I think the manuscript has a number of issues that require revision or clarification by the authors before further consideration for publication.

L50-53, artificially induced 2n gametes by chromosomes doubling of male or female gametes, and then obtained triploid through traditional cross-pollination. This is a more efficient and convenient method, and has achieved great success on poplar, Eucalyptus, Eucommia, jujube, rubber tree, etc. In addition to the three methods summarized, this method can be supplemented to make the statement more complete. Research team are also encouraged to try more methods on triploid breeding.

L84-85, Excellent work for seven years of consistent field experiments and management. I wonder whether the research team recorded the vegetative growth status of each plant. How did the identified triploid plants perform in the past seven years?

Fig 1, If possible, add a scale or ruler.

Table 1, missing column (tp) in table?

L152-155 and Fig 2, although the use of flow cytometry to identify plant ploidy is very mature and common, as this is the first time to induce triploid plants in date palm, it is strongly recommended to supplement the photos of chromosome counts in each triploids.

Table 2, e.g. 5/2 means February 5th? Please clarify.

Fig 4G-J, Fig 6E-F, Fig 7C-D, Fig 8, Fig 9, etc., if possible, add scale bars or rulers where necessary.

L241-243, endosperm balance number (EBN) may be an important reason for the low triploid yield, and a more in-depth discussion can be conducted based on this. Are obvious embryo or endosperm abortions observed in seeds that failed to become seedlings? The authors can also make suggestions for further triploid induction of date palm to make a more in-depth discussion of the manuscript.

L258, it is very interested to know the difference in fruit production ratio between T4 and diploid.

Conclusion, as mentioned previously, it is expected to provide suggestions for next steps in polyploid breeding plans for date palms.

Author Response

We would like to thank the academic editor and reviewers for their careful reading and insightful comments. This significantly helped us to improve our manuscript. In light of the comments and suggestions, we have thoroughly revised our proposal and we enclose the final version. Point-by-point responses to the comments are listed below. Modifications in the manuscript were highlighted in yellow.

Date palm is an important oasis ecological species and an important sugar fruit plant. In view of its biological and economic attributes, it is appropriate and effective to adopt a genetic improvement strategy of polyploid breeding. It is pleased to see that the research team has achieved good results, and hope that through continuous efforts, excellent varieties can be bred.

Overall, triploid breeding is successful for date palms, but what remains unresolved is, the yield of triploids produced by crossing tetraploids and diploids is very low, and the reproductive biology mechanism has not yet been fully revealed. This manuscript also does not essentially summarize methods for efficient breeding of triploid date palms. But what is beneficial is that the manuscript has characterized the fertility of triploid date palm in detail, which is very important for further developing date palm genetic improvement plans.

I think the manuscript has a number of issues that require revision or clarification by the authors before further consideration for publication.

L50-53, artificially induced 2n gametes by chromosomes doubling of male or female gametes, and then obtained triploid through traditional cross-pollination. This is a more efficient and convenient method, and has achieved great success on poplar, Eucalyptus, Eucommia, jujube, rubber tree, etc. In addition to the three methods summarized, this method can be supplemented to make the statement more complete. Research team are also encouraged to try more methods on triploid breeding.

A: We indeed neglected this valuable option. It was added as well as a recent reference.

L84-85, Excellent work for seven years of consistent field experiments and management. I wonder whether the research team recorded the vegetative growth status of each plant. How did the identified triploid plants perform in the past seven years?

A: No significant difference was detected in length or width growth between triploid and ordinary Deglet Noor palms. It is rather the vigor of the organs that differentiates the two types of genotype. We added this to the discussion : Although the triploid gave a more vigorous appearance, no difference was observed in overall plant length or width between the triploids and diploids. This was added to the manuscript.

Fig 1, If possible, add a scale or ruler.

A: Scale bars were added

Table 1, missing column (tp) in table?

A: The variable “tp” has been removed both from the table and the table legend.

L152-155 and Fig 2, although the use of flow cytometry to identify plant ploidy is very mature and common, as this is the first time to induce triploid plants in date palm, it is strongly recommended to supplement the photos of chromosome counts in each triploids. A: Karyotyping would be indeed interesting, but we cannot do it within a reasonable time frame. The triploids are still in the open field and we cannot harvest undamaged root tips with cells in meta-phase. This will be possible later, when off-shoots will be potted.

Table 2, e.g. 5/2 means February 5th? Please clarify.

A: this was clarified. 5/2 was for instance replaced by 5 Feb.

Fig 4G-J, Fig 6E-F, Fig 7C-D, Fig 8, Fig 9, etc., if possible, add scale bars or rulers where necessary.

A: Scale bars were added where possible

L241-243, endosperm balance number (EBN) may be an important reason for the low triploid yield, and a more in-depth discussion can be conducted based on this. Are obvious embryo or endosperm abortions observed in seeds that failed to become seedlings? The authors can also make suggestions for further triploid induction of date palm to make a more in-depth discussion of the manuscript.

A: Thank you for your remark. We added following sentence to the discussion:  “According to the concept of Endosperm Balance Number (EBN), that states that the hybrid endosperm needs a 2:1 maternal to paternal ratio for normal development, this cross has a problematic 4:1 ratio, which may also explain the high number of failing offspring” and added a recent reference:

Li H.; Gan J; Xiong H; Mao X; Li S; Zhang H; Liu C.; Hu G.; Fu, J. Production of triploid germplasm by inducing 2n pollen in Longan. Horticulturae, 2022, 8(5), pp; 437. https://doi.org/10.3390/horticulturae8050437

L258, it is very interested to know the difference in fruit production ratio between T4 and diploid.

A: We added these data to the table 3 and added it to the discussion.

Conclusion, as mentioned previously, it is expected to provide suggestions for next steps in polyploid breeding plans for date palms.

A: This was done

Reviewer 2 Report

Comments and Suggestions for Authors

The aim of the manuscript is the evaluation of the potential of four specimens of triploid date palms. There are few publications on triploidy in Phoenix dactylifera, for this reason the research carried out here is very interesting. The manuscript is clearly laid out, which makes it easy to read.

 I would like to underline some aspects that would improve the manuscript.

 It would be interesting to compare in a table the production of edible date fruit of a Deglet Nour diploid palm, of a Deglet Nour tetraploid palm and of the T3 and T4 triploid palms (they can give the value in weight or in number of date fruits), as well as to evaluate the size and organoleptic quality of the date fruits.

 -       What type of male palm is the P3 cultivar?

-       Could you insert a photo of the pollen of the P3 cultivar?

-       P3 is the cultivar that is generally used to tap the palms of the oasis of Tozeur?

Table 1. Characteristics of fruits produced by the tetraploid ‘Deglet Nour’ shoot

 -per bunch or per tetraploid ‘Deglet Nour’ shoot? How many bunches does the tetraploid palm have?

-Column “%sf” change “sf%”

- tp: triploid plants (table 1 caption). What does it mean? There is no column tp

Table caption 3. Fruit weight, seed weight. It is necessary to indicate the weight units.

Figure caption 5 and 6. The letters are lowercase and they should be uppercase letters.

Figure 6. The image B has two number 1, it must be 1 and 2 (b) flower axis of T3 (1) and ‘Deglet Nour’ (2).

Figure caption 7. “(d) flowers of ‘Deglet Nour’ (top)” must be “(d) flowers of ‘Deglet Nour’ (top), flowers of T4 bottom”.

Figure caption 7.(e) fruits of T3 in rutab stage”, is correct? Perhaps may be (e) fruits of T4 in rutab stage”.

Paragraph line 274-284 is similar to paragraph line 285-297, the authors should delete one of the paragraphs.

You should explain better why T3 is preferable to T4. T3 fruit is bigger than T4 and T3 seed is smaller than T4.

Are the fruits of T3 better quality than those of T4? Do you know the organoleptic characteristics of T3 and T4? Do they have a similar longer shelf life?

Author Response

We would like to thank the academic editor and reviewers for their careful reading and insightful comments. This significantly helped us to improve our manuscript. In light of the comments and suggestions, we have thoroughly revised our proposal and we enclose the final version. Point-by-point responses to the comments are listed below. Modifications in the manuscript were highlighted in yellow.

The aim of the manuscript is the evaluation of the potential of four specimens of triploid date palms. There are few publications on triploidy in Phoenix dactylifera, for this reason the research carried out here is very interesting. The manuscript is clearly laid out, which makes it easy to read. I would like to underline some aspects that would improve the manuscript.

It would be interesting to compare in a table the production of edible date fruit of a Deglet Nour diploid palm, of a Deglet Nour tetraploid palm and of the T3 and T4 triploid palms (they can give the value in weight or in number of date fruits), as well as to evaluate the size and organoleptic quality of the date fruits.

A: These data were to table 3 and were further discussed.

What type of male palm is the P3 cultivar?

A: A common diploid pollinator, we added this clarification in M&M

Could you insert a photo of the pollen of the P3 cultivar?

A: We inserted such a photo in Fig 4J

P3 is the cultivar that is generally used to tap the palms of the oasis of Tozeur? C’est un pollinisateur parmi quelques milliers d’autres utilizes par les agriculteurs pour pollinizer les palmiers femelles

Table 1. Characteristics of fruits produced by the tetraploid ‘Deglet Nour’ shoot -per bunch or per tetraploid ‘Deglet Nour’ shoot? How many bunches does the tetraploid palm have?

A: We only have data for the tetraploid Deglet Nour offshoot. The number of bunches vary depending on the year, but it is around 4-6.

Column “%sf” change “sf%”

A: This was corrected

- tp: triploid plants (table 1 caption). What does it mean? There is no column tp. OK, I omit it

A: The variable “tp” has been removed both from the table and the table legend.

Table caption 3. Fruit weight, seed weight. It is necessary to indicate the weight units.

A: This was corrected

Figure caption 5 and 6. The letters are lowercase and they should be uppercase letters.

A: This was corrected

Figure 6. The image B has two number 1, it must be 1 and 2 (b) flower axis of T3 (1) and ‘Deglet Nour’ (2).

A: This was corrected

Figure caption 7. “(d) flowers of ‘Deglet Nour’ (top)” must be “(d) flowers of ‘Deglet Nour’ (top), flowers of T4 bottom”.

A: This was corrected

Figure caption 7. “(e) fruits of T3 in rutab stage”, is correct? Perhaps may be “(e) fruits of T4 in rutab stage”.

A: This was corrected

Paragraph line 274-284 is similar to paragraph line 285-297, the authors should delete one of the paragraphs.

A: This part was deleted

You should explain better why T3 is preferable to T4.

A: We explained this better.  T3 fruit is bigger than T4 and T3 seed is smaller than T4. T3 has at the same time higher seed weight and less ratio (seed weight/ total weight) compared with T4.

Are the fruits of T3 better quality than those of T4? Do you know the organoleptic characteristics of T3 and T4?

A: We explained better: T3 had a superior taste and texture. This lucky shot shows that triploids can be fantastic in this regards.

Do they have a similar longer shelf life?

A: The plants just started fruit production but every autumn the harvest increases. We studied only the physical propriety of fruits and we will analyze their chemical properties next season. We added this to the conclusion.

Round 2

Reviewer 1 Report

Comments and Suggestions for Authors

Maybe there is a problem with the layer display. Please do not cover the subject with the scale bar. Place the scale bar in Figures 4 and 5 in the corner of the picture.

Please add scale bar to Figures 8 and 9.

Since this is the first report of triploid breeding in date palms, I still strongly recommend the authors add photos of triploid chromosome counts in some way in the future.

Overall, as an important polyploid breeding achievement, I think the revised manuscript is ready for publication.

Author Response

Thank you for your attentiveness.

Missing scale bars were added and they were integrated the scalebars into the figures so there was no problem with the layer display anymore. Please do not cover the subject with the scale bar.